# PFDN4 as a Prognostic Marker Was Associated with Chemotherapy Resistance through CREBP1/AURKA Pathway in Triple-Negative Breast Cancer

**DOI:** 10.3390/ijms25073906

**Published:** 2024-03-31

**Authors:** Shih-Ho Wang, Cheng-Hsi Yeh, Chia-Wei Wu, Chia-Yi Hsu, Eing-Mei Tsai, Chao-Ming Hung, Yi-Wen Wang, Tsung-Hua Hsieh

**Affiliations:** 1Division of General Surgery, Kaohsiung Chang Gung Memorial Hospital, Chang Gung University College of Medicine, Kaohsiung 83301, Taiwan; 2Department of Medical Research, E-Da Hospital/E-Da Cancer Hospital, I-Shou University, Kaohsiung 82445, Taiwan; snoopy79101@gmail.com (C.-W.W.); ywwang0228@gmail.com (Y.-W.W.); 3Department of Obstetrics and Gynecology, Kaohsiung Medical University Hospital, Kaohsiung Medical University, Kaohsiung 80756, Taiwan; husonweihsu@hotmail.com (C.-Y.H.);; 4Department of Surgery, E-Da Cancer Hospital, I-Shou University, Kaohsiung 82445, Taiwan

**Keywords:** PFDN4, chemotherapy resistance, triple-negative breast cancer

## Abstract

Breast cancer is the most common malignancy and its incidence is increasing. It is currently mainly treated by clinical chemotherapy, but chemoresistance remains poorly understood. Prefolded proteins 4 (PFDN4) are molecular chaperone complexes that bind to newly synthesized polypeptides and allow them to fold correctly to stabilize protein formation. This study aimed to investigate the role of PFDN4 in chemotherapy resistance in breast cancer. Our study found that PFDN4 was highly expressed in breast cancer compared to normal tissues and was statistically significantly associated with stage, nodal status, subclasses (luminal, HER2 positive and triple negative), triple-negative subtype and disease-specific survival by TCGA database analysis. CRISPR knockout of PFDN4 inhibited the growth of 89% of breast cancer cell lines, and the triple-negative cell line exhibited a stronger inhibitory effect than the non-triple-negative cell line. High PFDN4 expression was associated with poor overall survival in chemotherapy and resistance to doxorubicin and paclitaxel through the CREBP1/AURKA pathway in the triple-negative MDAMB231 cell line. This study provides insightful evidence for the value of PFDN4 in poor prognosis and chemotherapy resistance in breast cancer patients.

## 1. Introduction

Breast cancer is one of the most commonly diagnosed cancers in women and is the second most common cancer killer in women after lung cancer. With advances in technology and healthcare, the cure rate for breast cancer has increased over the years. In terms of treatment, surgical removal is followed by chemotherapy. Currently, neoadjuvant chemotherapies as a treatment option in breast cancer, including anthracyclines and paclitaxel, are used as systemic primary therapies in chemotherapy. The chemotherapeutic effect of anthracyclines is to induce DNA double-strand breaks (DSBs), leading to apoptosis [1] and paclitaxel binds to microtubules and causes assembly of nonfunctional microtubules. It is worth noting that the ability to repair DNA damage varies between breast cancer subtypes and is closely related to chemotherapy resistance. However, the overall survival of triple-negative breast cancer is not always improved compared to other breast cancer subtypes [2]. Chemotherapy resistance is a major obstacle to the success of cancer treatment, especially in patients with metastatic cancer, where about 90% of failures are due to chemotherapy resistance [3]. Prefolding protein subunit (PFDN) consists of six different subunits (PFDN1-6), two α subunits (PFDN3 and 5) and four β subunits (PFDN1, 2, 4 and 6) [4] and is a hexameric cofacilitator complex that helps regulate the monomeric folding of actin and tubulin [5]. Previous studies have found that all six PFDN subunits are involved in cancer-related biology [6,7,8,9]. Among them, PFDN3 and PFDN5 subunits have been shown to form complexes with many proteins [10,11]. However, there is still much to be clarified in the study of β subunits. The expression of PFDN4, also known as C-1, is a transcription factor and has the ability to regulate the cell cycle [12,13] and may be closely associated with the occurrence, development and poor prognosis of several tumors, including hepatocellular carcinoma and colorectal cancer [14]. Aurora kinase A (AURKA) is a serine/threonine cell cycle kinase that is aberrantly expressed and over-amplified in many cancers, including colorectal, gastric and pancreatic cancers [15], leading to genetic mutations, morphological changes and promoting carcinogenesis. It also plays an important role in mitosis during the cell cycle and in regulating cell proliferation [16]. In this study, we analyzed the role of PFDN4 in breast cancer using the TCGA database and investigated the effect of chemotherapeutic agents via the CREBP1/AURKA pathway in triple-negative breast cancer, leading to the recent discovery that PFDN4 could serve as a potential biomarker for prognosis and chemotherapy resistance.

## 2. Results

### 2.1. Association of PFDN with Disease-Specific Survival and Clinicopathological Features in Breast Cancer Subjects

First, we evaluated the disease-specific prognostic value of the PFDN family including PFDN1, 2, 4 and 6 by TCGA PanCancer Atlas using Kaplan–Meier analysis of cBioPortal database (https://www.cbioportal.org, accessed on 1 January 2024) in breast invasive carcinoma (Figure 1A–D). The PFDN4 gene samples were divided into two groups using the median performance of the gene (high vs. low expression). The results show that high expression of PFDN4 was significantly associated with survival rate (*p* = 0.0235) in breast invasive carcinoma. Next, the expression of PFDN4 across TCGA pan-cancer was shown in Figure 1E (BRCA: breast invasive carcinoma). It was found that PFDN4 is highly expressed in most tumors relative to normal tissue, and the result also indicated that the expression level of PFDN4 was higher in BRCA (n = 1097) compared to its matched normal tissues (n = 114) (Figure 1F) through UALCAN database (https://ualcan.path.uab.edu, accessed on 1 January 2024). In addition, high expression of PFDN4 was associated with stages 1, 2 and 3 (Figure 1G) and nodal metastasis status (N0, N1 and N2) (Figure 1H). These results found that high expression of PFDN4 was a prognostic factor and associated with tumor stage and nodal metastasis.

### 2.2. PFDN4 Mediates Cell Growth and Cell Cycle in Triple-Negative Breast Cancer Cell Lines

We also examined the expression of PFDN4 in breast cancer based on breast cancer subclasses including luminal, HER2 positive and triple-negative group. The results showed that the expression of PFDN4 was significantly associated with triple negative compared to normal (Figure 2A, *p* < 0.01). We also used CRISPR knockout screens of the PFDN4 system to analyze cell growth in 12 non-triple-negative and 18 triple-negative breast cancer cell lines. The results showed that silencing PFDN4 has a greater ability to inhibit cell growth in triple-negative compared to non-triple-negative breast cancer cell lines (Figure 2B,C, *p* = 0.039). We also examined the cell cycle by flow cytometry in the breast cancer cell lines MCF7 (non-triple-negative) and MDAMB231 (triple-negative). The results showed that PFDN4 overexpression decreased the percentage of cells in G1/M phase, but increased S phase in MCF7 (Figure 2D). In contrast, PFDN4 knockdown decreased the percentage of cells in S phase (Figure 2E). MCF7 and MDAMB231 were no difference between control and PFDN4 in the percentage of cells in G2/M. This result suggests that PFDN4 may play an important role in triple-negative breast cancer with high frequency of cell growth and cell cycles.

### 2.3. Role of PFDN4 in Chemotherapy Resistance

We further investigated the ability of PFDN4 using cBioPortal database to regulate chemotherapy resistance. The disease-specific prognostic value of PFDN4 (PFDN4 uses median to differentiate between high and low expression) with drug including doxorubicin (Figure 3A), paclitaxel (Figure 3B), cyclophosphamide (Figure 3C), tamoxifen (Figure 3D), anastrazole (Figure 3E) and docetaxel (Figure 3F) was determined using Kaplan–Meier analysis in breast invasive carcinoma. The results showed that high expression of PFDN4 with doxorubicin (*p* = 0.0003701), paclitaxel (*p* = 0.001294) and cyclophosphamide (*p* = 0.003080) was significantly associated with survival rate in breast invasive carcinoma, but the other drugs were not. Doxorubicin, paclitaxel and cyclophosphamide do not improve survival in breast cancer patients with high PFDN4 expression. These results suggest that PFDN4 has an antagonistic effect on doxorubicin, paclitaxel and cyclophosphamide in invasive breast cancer.

### 2.4. PFDN4 Antagonizes the Effects of Doxorubicin and Paclitaxel in Breast Cancer Cell Lines

We further investigated the effects of PFDN4 on chemotherapy resistance to doxorubicin and paclitaxel in breast cancer cell lines. We used MCF7 (non-triple-negative) and MDAMB231 (triple-negative) breast cancer cell lines to perform chemotherapy resistance experiments and verify the expression of PFDN4 in MCF7 and MDAMB231 cells. The results showed that the level of PFDN4 in MDAMB231 cells is higher than that in MCF7 (Figure 4A). We also used a PFDN4 overexpression plasmid and siRNA-1/2 to detect cell growth by CCK-8 assay with doxorubicin and paclitaxel treatment in MCF7 and MDAMB231 cell lines. The results showed that PFDN4 overexpression increased cell growth in MCF7 (Figure 4B), but PFDN4 siRNA1/2 decreased cell growth in MDAMB231 (Figure 4C). We also analyzed the effect of doxorubicin and paclitaxel on cell growth in MCF7 and MDAMB231. The results show that doxorubicin and paclitaxel have greater ability to inhibit cell growth in MCF7 than MDAMB231 breast cancer cell lines (Figure 4D,E), and PFDN4 overexpression significantly inhibited the effect of doxorubicin and paclitaxel, but siRNA knockdown of PFDN4 expression enhanced the effect of doxorubicin and paclitaxel in cells (Figure 4F,G). These results showed that PFDN4 has the ability to regulate resistance to doxorubicin and paclitaxel chemotherapy.

### 2.5. PFDN4 Regulates Breast Cancer Chemotherapy Resistance through AURKA

We also analyzed the gene correlated with PFDN4 in chemotherapy resistance. Using cBioPortal and UALCAN, we found that AURKA was positively correlated (R = 0.558) and significantly correlated (*p* = 2.64 × 10^−82^) with PFDN4 (Figure 5A,B). A previous study found that PFDN4 binds to CREBP1 [17] and transcription factor affinity prediction (TRAP) shows that CREBP1 is an upstream transcription factor of AURKA (Figure 5C). CREB is a basic region transcription factor because it possesses a leucine zipper domain (bZIP) that promotes dimerization and mediates its binding to DNA. Among them, CREBP1 (also known as ATF-2) is a highly homologous protein and also has a bZIP structure [18]. As a first step, we investigated whether PFDN4 has the ability to regulate AURKA expression. We found that PFDN4 overexpression induced the level of ARUKA, but PFDN4 siRNA 1/2 decreased the expression of AURKA (Figure 5D,E). Our results also showed that PFDN4 bound to CREBP1 and CREBP1 complexed with AURKA promoter region by CH-IP assay in MCF7 and MDAMB231 (Figure 5F,G). Next, we used a luciferase assay (pGL3) to verify whether CREBP1 binds to the AURKA promoter region. We constructed the pGL3 wild type (WT) and mutated (MT) fragments of the AURKA 5′-UTR and co-transfected with CREBP1 into 293T cells. The results showed that CREBP1 overexpressed increased the activity of the luciferase reporter in a dose-dependent manner in the WT group, but not in the MT group (Figure 5H,I). Finally, we investigated the effect of AURKA on chemotherapy resistance and found that AURKA overexpression could antagonize the toxicity of doxorubicin and paclitaxel drugs, and AURKA siRNA 1/2 could enhance the antitumor activity of paclitaxel and doxorubicin drugs in human breast cancer cell lines (Figure 5J,K).

## 3. Discussion

Each year, a large number of patients lose their lives due to chemotherapy failure in breast cancer. In our study, we present evidence that PFDN4 overexpression is associated with a higher incidence of chemoresistance in triple-negative breast cancer than in non-triple-negative breast cancer, and is also a prognostic factor for poor prognosis in patients receiving chemotherapy. Doxorubicin and paclitaxel are widely used chemotherapy drugs for breast cancer patients that can effectively reduce the risk of recurrence and mortality in breast cancer patients [19,20]. Unfortunately, when cancer cells are cumulatively exposed to certain doses of paclitaxel and doxorubicin, the cancer cells develop a multidrug-resistant phenotype, and the doxorubicin-resistant cells even show significant cross-resistance to other chemotherapeutic drugs [21,22]. Among the subtypes of drug-resistant breast cancer, triple-negative breast cancer is the most common, and doxorubicin and paclitaxel have poorer inhibition rates in triple-negative breast cancer than in non-triple-negative breast cancer [23], and the reasons for this are not yet clear. The PFDN family has also been linked to drug resistance in previous studies. PFDN2 has the ability to regulate the folding of microtubule proteins, which when mutated or abnormal can lead to misfolding of microtubule proteins, and Taxel can inhibit cell division and slow the proliferation of cancer cells by stabilizing microtubules and destroying the mitotic spindle [24]. Therefore, PFDN2 has the ability to regulate the development of resistance to Taxel treatment in breast cancer. PFDN4 has the ability to antagonize mitotic spindle agents (paclitaxel), so we suggest that PFDN4 has the ability to affect mitotic spindle. In the future, we will further explore how PFDN4 regulates the mitotic spindle in breast cancer. 

Cyclophosphamide is an alkylating agent that can alkylate DNA guanine and form bonds with double-stranded DNA, making cellular DNA irreparable [25]. Doxorubicin also has the ability to rapidly inhibit mitotic activity and inhibit DNA nucleic acid synthesis [26]. In our study, PFDN4 showed higher expression in breast cancer than in normal tissue, and in triple-negative breast cancer than in non-triple-negative breast cancer. We found that PFDN4 increases the S phase of the cell cycle and affects the activity of drugs through mediating cell cycle, which provides an effective mechanism for verifying drug resistance and novelty. Previous studies have also found that PFDN4 is highly expressed in breast cancer cell lines and identified as an oncogenic gene [12], this is consistent with our current research. Our research further found that PFDN4 has the ability to antagonize chemotherapy drugs in breast cancer. In addition, a study conducted by Miyoshi et al. in colorectal cancer found that high expression of PFDN4 may be an indicator of relatively good prognosis [14]. These findings suggest that PFDN4 may play different roles in different tumors. 

The PFDN family has been shown to function as a cofactor in the regulation of cytoskeletal rearrangements, with major studies focusing on epithelial-mesenchymal transition (EMT) and cancer progression. Studies of PFDN1 in colorectal cancer have shown that PFDN1 is highly expressed in colorectal cancer compared to neighboring normal tissues. Silencing of PFDN1 resulted in G2/M cell cycle arrest and cytoskeletal defects in colorectal cancer, leading to significant inhibition of cell proliferation and motility [27]. In addition, PFDN2 increases cell cycle progression through the hnRNPD-MYBL2 pathway and may act as a potential biomarker in gastric cancer [28]. PFDN5 can activate c-MYC transcription factor and regulate cell cycle in human HeLa cells [29]. PFDN6 was overexpressed in human glioma tissues and this significantly correlated with a poor survival rate. PFDN6 knockdown blocked cell cycle progression and inhibited cell proliferation and migration [30]. These previous studies have suggested that the PFDN family has the ability to regulate the cell cycle. Our experimental results also found that PFDN4 has high performance in MDAMB231 (mesenchymal type) compared to MCF7 (epithelial type). After silencing of PFDN4, the ability of breast cancer cells in cell growth and motility is reduced. This has the same biological function as PFDN1, PFDN2, PFDN5 and PFDN6. 

Previous studies have shown that overexpression of AURKA affects resistance to paclitaxel and cisplatin, which in turn reduces the activity of chemotherapeutic drugs [31]. The role of AURKA is participation in the regulation of the spindle during early mitosis. Mitotic defects in many human cancers are primarily due to abnormal expression of AURKA. Some AURKA substrates are involved in important oncogenic signaling including with GSK-3β, β-catenin, Twist, ERα, IκBα, and YAP [32,33,34]. In addition, AURKA interacts with ERα and phosphorylates at Ser167 and Ser305 to promote increased transcriptional activity of cyclin D1. Interestingly, high-performing AURKA was associated with poorer survival in patients with ERα-positive tumors [35]. Hayakawa et al. also found that CREBP1 can stimulate resistance to DNA damaging agents by activating DNA repair and regulating chemotherapy resistance mechanisms [36]. Interestingly, the growth inhibition effect of tamoxifen on MCF7 cell growth is reduced when CREBP1 is silenced. These results indicate that CREBP1 has better tumor suppressive ability through tamoxifen treatment in ER-positive breast cancer [37]. It suggests that breast cancer patients with high AURKA or CREBP1 performance should be more likely to receive hormonal therapy. Our results found that PFDN4/CREBP1/AURKA is a drug resistance signaling pathway of doxorubicin, paclitaxel, cyclophosphamide. Therefore, we suggest that breast cancer patients with high PFDN4 expression should receive tamoxifen treatment. 

Our experimental results also showed that PFDN4 induced cell growth and metastasis via CREBP1/AURKA and resisted inhibition of cell death by doxorubicin and paclitaxel.

CREBP1 is also involved in regulating cell growth and proliferation in cancer cells. CREBP1 can be activated through the JNK signaling pathway to promote cell proliferation and lead to hepatocellular carcinoma [38]. In renal cell carcinoma, CREBP1 was also found to be highly expressed in metastatic tumors and correlated with aggressiveness and poor prognosis in clinicopathological features [39]. The levels of AURKA were significantly higher in bladder cancer compare to normal tissues. High expression of AURKA was strongly associated with stage, grade and poor survival rates. Silencing AURKA inhibits cell proliferation in the bladder cancer cell lines. Therefore, we believe that PFDN4, CREBP1 and AURKA play an important role in the process of chemotherapy resistance by antagonizing the DNA repair mechanism. 

Currently, about 5.6% (60/1003 (low expression) + 60 (high expression))) of breast cancer patients are from the PFDN4-high expression group. Our study suggested that PFDN4 high-expressing groups should reduce the use of mitotic spindle, alkylating agents for drug treatment, instead using tamoxifen or anastrazole drugs to improve healing effects. In addition, PFDN4 inhibitors or other relatively α- and β-subunit-selective PFDN inhibitors can also be used in future clinical drug development. However, the details of the antagonistic mechanism still need to be further investigated and studied in detail, which will help to develop novel and effective therapeutic strategies for breast cancer.

## 4. Materials and Methods

### 4.1. Cell Lines

Human breast cancer cell lines (MCF7, non-triple-negative and MDAMB231, triple-negative) were purchased from the American Type Culture Collection (ATCC) and grown at 37 °C in a 5% CO_2_ atmosphere in DMEM (GIBCO, Waltham, MA, USA) supplemented with 10% fetal bovine serum (FBS) and 1% penicillin/streptomycin (GIBCO, USA). The medium is changed every two days.

### 4.2. Transfection 

Plasmid and siRNA transfections were performed using TurboFect Transfection Reagent (Thermo Scientific, Waltham, MA, USA) according to the manufacturer’s protocol. The pCMV6-PFDN4 (RC203137) and CREBP1 (RC218983) siRNAs were obtained from ORIGENE. The target sequences were as follows: si-PFDN4-1: CAT TCT CAA GAA GAA ACG CAA; si-PFDN4-2: CGG AAT ACA AGT AGA ATC ACA; si-AURKA-1: AGG CCA CTG AAT AAC ACC CAA; si-AURKA-2: GAG TCT ACC TAA TTC TGG AAT.

### 4.3. Cell Growth

A total of 5 × 10^3^ cells were seeded in 96-well plates after transfection and incubated 24 h later into a cell growth assay using Cell Count Kit-8 (CCK-8, Dojindo, Kumamoto, Japan). Absorbance (450 nm) was detected in each sample using a microplate reader (Bio-Rad, Hercules, CA, USA). 

### 4.4. QPCR and Chromatin Immunoprecipitation Quantitative Real-Time PCR

To confirm the intracellular expression of PFDN4 and AURKA, mRNA was determined by reverse transcription PCR (RT-PCR). Cellular RNA was extracted with TRIzol (Invitrogen) and transcribed into cDNA using a reverse transcription system kit (Promega, Madison, WI, USA). For chromatin immunoprecipitation quantitative real-time PCR, PFDN4 was overexpressed in the cell and the proteins bound to PFDN4 were precipitated using immunoprecipitation technology. Finally, DNA fragments were obtained and subjected to RT-PCR, which was performed using SYBR Green PCR Master Mix (Applied Biosystems, Foster City, CA, USA) in an ABI 7500 real-time PCR system (Applied Biosystems). Primers used in the experiments were PFDN4-F-5′-CGG TAG TCC AGT CCC AAG ATG-3′,PFDN4-R-5′-TCA GCT CTG TGA TTC TAC TTG TAT TCC-3′ [14]; AURKA-F-5′-AGT TGG AGG TCC AAA ACG TG-′3, AURKA-R-5′-TCC AAG TGG TGC ATA TTC CA-′3 [40]; CREBP1 XRE-F-5′-GGA CTT TAT ACT CCA ATG CGT C-3′, CREBP1 XRE-R-5′-CCG GTA AAT AGC CAC TTC G-3′.

### 4.5. UALCAN

UALCAN is a comprehensive web-based resource for analyzing extensive cancer data, providing users with graphs and plots of gene performance and survival curves, as well as evaluating promoter DNA methylation information and performing pan-carcinogenic gene performance analyses [41]. In this study, we used UALCAN data to investigate the role of the PFDN4 gene among 24 human cancer subtypes. In addition, we also used UALCAN platform to analyze the correlation between PFDN4 expression and breast cancer patients with different clinicopathological characteristics. The expression of PFDN4 gene was measured as transcripts per million (TPM) reads and expression differences between the medians of the normal and cancer groups were compared by Student’s *t* test. A *p* value < 0.05 was used to indicate a significant score.

### 4.6. cBioPortal

The cBioPortal Cancer Genome Database is an open-access web platform for interactive exploration of cancer genomic data among many cancer genomes [42]. We selected cancer study “Breast Invasive Carcinoma (TCGA, PanCancer Atlas)” and PFDN4 for our study as we obtained data from an open access database. We analysed the survival rate of PFDN4 in breast cancer and compared the survival rate after clinical drug treatment.

### 4.7. Gene Effect Scores

Gene effect scores were released through Broad’s Achilles and Sanger’s SCORE projects, which are mainly big data values calculated through CRISPR knockout screening and combined with cancer genome ecology calculations. Negative scores represent cell growth inhibition and/or death after gene knockout, indicating that the gene may be required in a specific cell line. A score of 0 or above means that the gene is not an essential gene [43,44].

### 4.8. Luciferase Report Assay

AURKA-WT-3′-UTR (TGACGTCA) and AURKA-MT-3′-UTR (TTAAGTTA) were constructed using a luciferase report vector (PGL3). Dose-dependent CREBP1 plasmid and luciferase reporter vector were co-transfected using TurboFect Transfection Reagent (Thermo Scientific) according to the manufacturer’s protocol. After 24 h, the level of activity (firefly/renilla) was analyzed using the Dual-Glo Luciferase Assay (Promega, Madison, WI, USA) according to the manufacturer’s protocol.

### 4.9. Flow Cytometry 

A total of 5 × 10^5^ cells were plated in 6-well plates. Twenty-four hours after transfection, the accumulated cells were digested with trypsin, then rinsed with cold PBS and incubated with propidium iodide (PI) staining for approximately 10 min at room temperature. Finally, the cells were washed twice with cold PBS and cell cycle was analyzed by flow cytometry (Guava, Luminex, TX, USA).

## Figures and Tables

**Figure 1 ijms-25-03906-f001:**
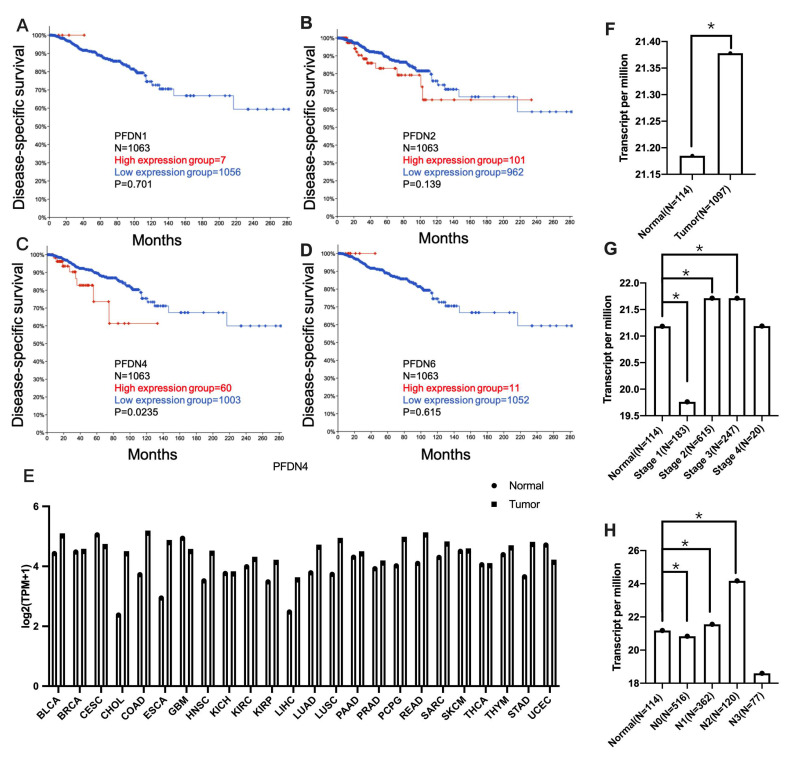
PFDN4 is a prognostic factor associated with survival in TCGA breast cancer. (**A**–**D**) Disease-specific survival plots of PFDN1, PFDN2, PFDN4 and PFDN6 in the TCGA breast cancer atlas. (**E**) Expression of PFDN4 in the TCGA pan-cancer view including tumor and normal group. Cancer names based on TCGA study abbreviation table. (**F**) Expression of PFDN4 in normal (N = 114) and tumor samples (N = 1097). (**G**) PFDN4 was associated with stage and (**H**) nodal status. *: *p* value < 0.05 was used to indicate a significant score.

**Figure 2 ijms-25-03906-f002:**
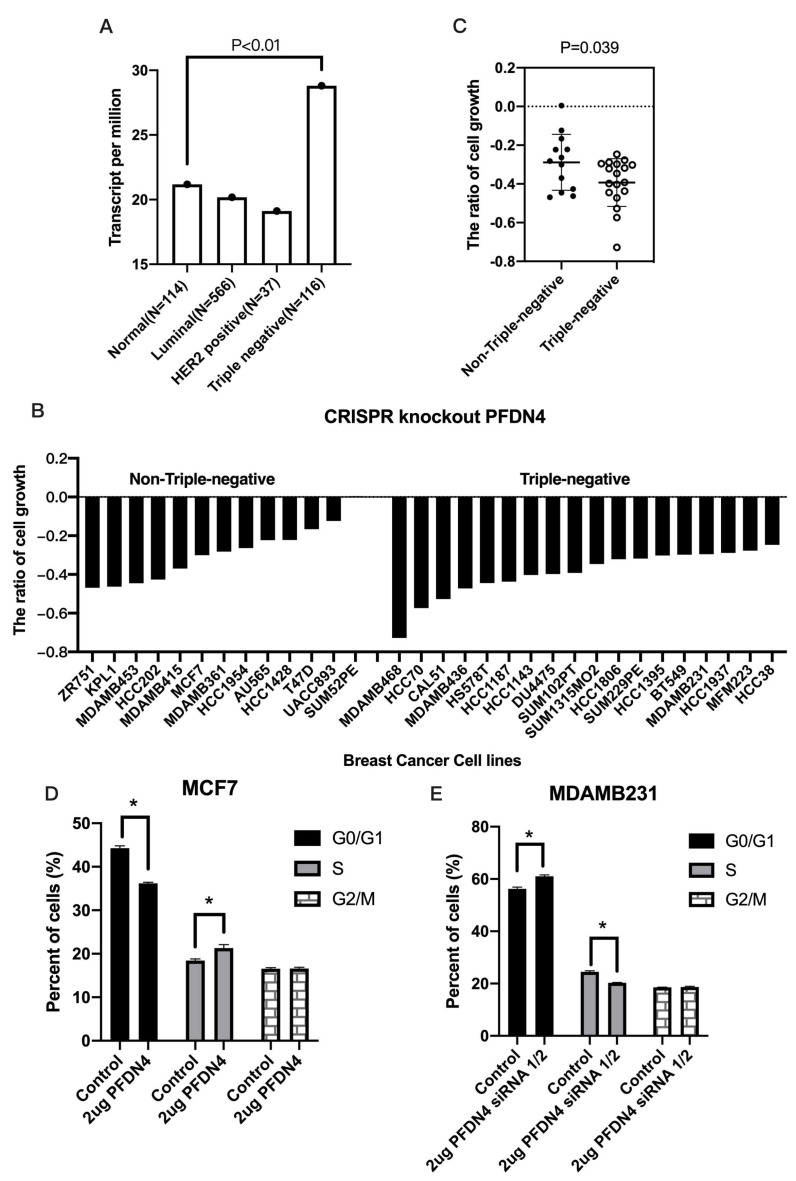
PFDN4 increased cell growth in triple negative breast cancer. (**A**) PFDN4 was associated with subclasses including luminal (N = 566), HER2 positive (N = 37) and triple-negative (N = 116). (**B**) Gene effect scores of PFDN4 genes in breast cancer cell lines, (**C**) Group of triple-negative and non-triple-negative breast cancer cell lines. Cell cycle was analyzed by flow cytometry in MCF7 (**D**) and MDAMB231. (**E**) Transfected with PFDN4 overexpression plasmid and siRNA-1/2, respectively. *: *p* value < 0.05 was used to indicate a significant score.

**Figure 3 ijms-25-03906-f003:**
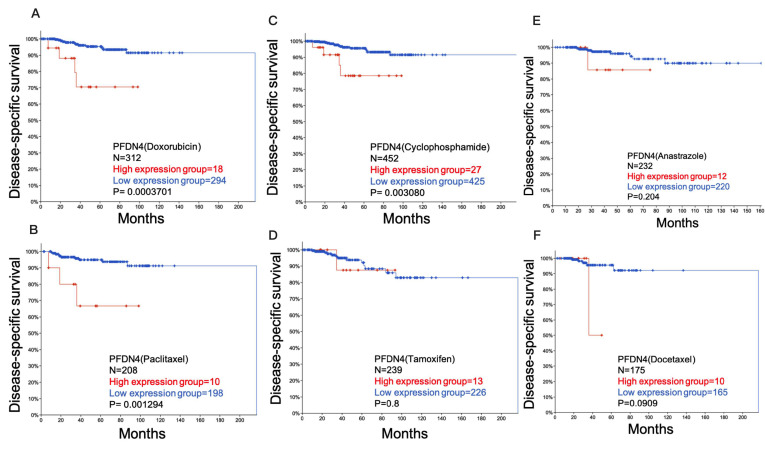
PFDN4 is a chemotherapy resistance factor associated with survival when treated with chemotherapeutic drugs in TCGA breast cancer. Disease-specific survival plots of PFDN4 treatment with (**A**) Doxorubicin, (**B**) Paclitaxel, (**C**) Cyclophosphamide, (**D**) Tamoxifen, (**E**) Anastrazole and (**F**) Docetaxel.

**Figure 4 ijms-25-03906-f004:**
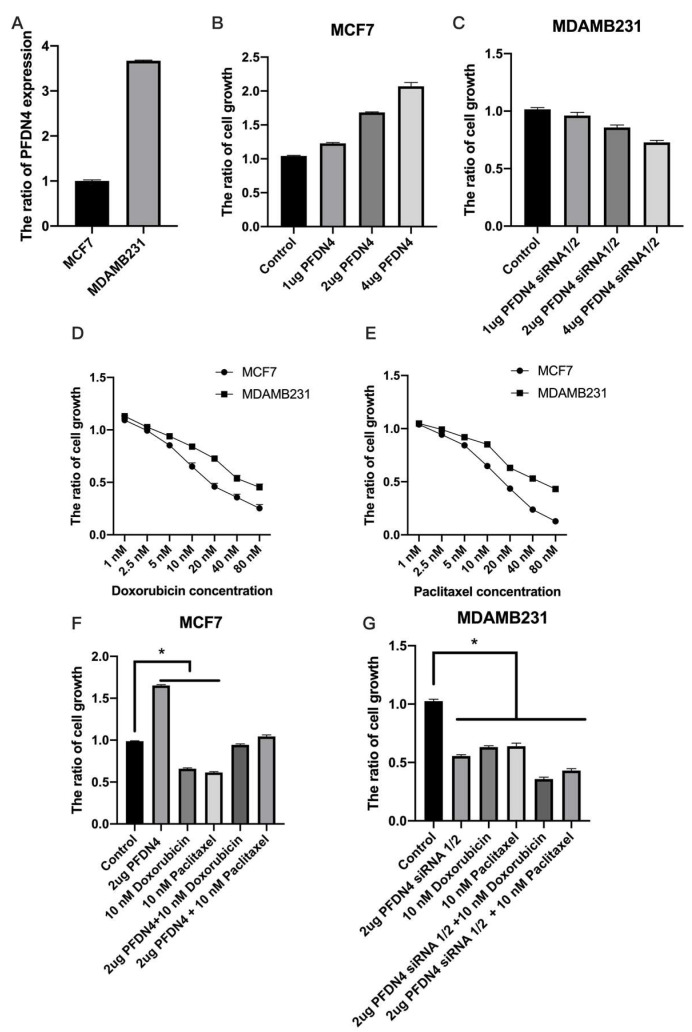
PFDN4 antagonizes paclitaxel and amygdalin to inhibit breast cancer cell growth. (**A**) The expression of PFDN4 was analyzed by RT-PCR in MCF7 and MDAMB231. The cell growth was analyzed in a dose-dependent manner in PFDN4 plasmid (**B**), siRNA-1/2 (**C**), Doxorubicin and (**D**) Paclitaxel (**E**) in MCF7 and MDAMB231, respectively. (**F**,**G**) The cell growth was analyzed in MCF7 and MDAMB231 when treated with 2 µg PFDN4, 2 µg PFDN4 siRNA-1/2, 10 µM paclitaxel, 10 µM doxorubicin, 2 µg PFDN4 + 10 µM paclitaxel, 2 µg PFDN4 + 10 µM doxorubicin, 2 µg PFDN4 siRNA-1/2 + 10 µM paclitaxel and 2 µg PFDN4 siRNA-1/2 + 10 µM doxorubicin. *: *p* value < 0.05 was used to indicate a significant score.

**Figure 5 ijms-25-03906-f005:**
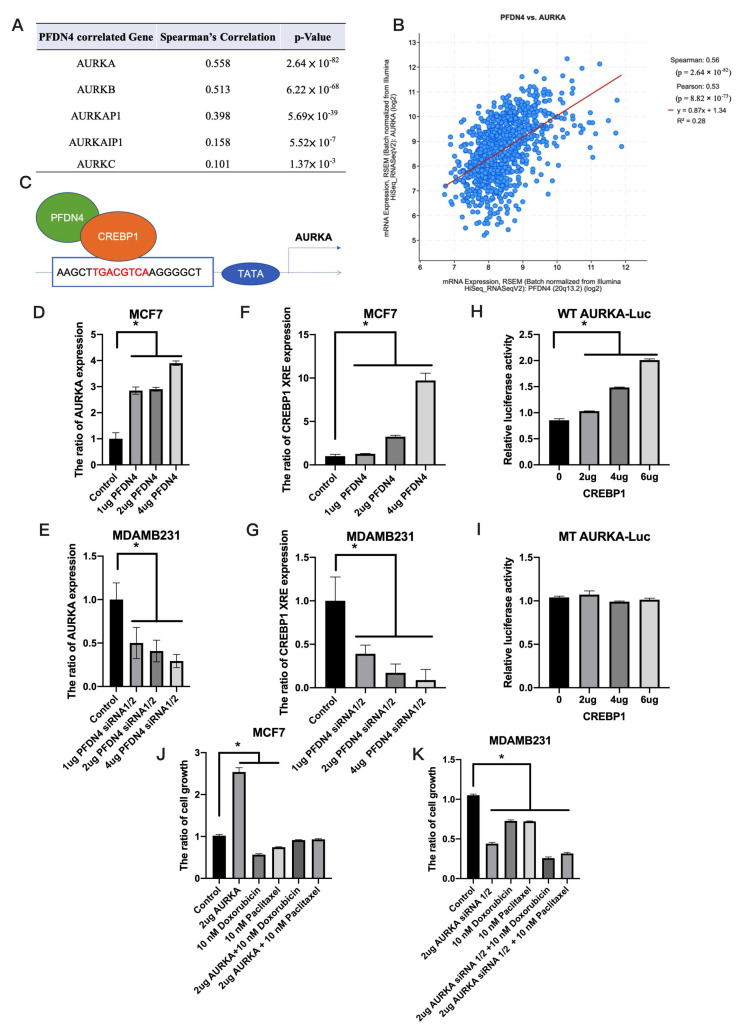
PFDN4 regulates the downstream transcription factor CREBP1 and AURKA to affect the ability of drug resistance. (**A**,**B**) AURKA, AURKB, AURKAP1, AURKAIP1 and AURKC were associated with PFDN4 in breast cancer. (**C**) Graphic showing that PFDN4 binds to the transcription factor CREBP1 and activates the AURKA promoter region. (**D**,**E**) The expression of AURKA was analyzed in MCF7 and MDAMB231 in a PFDN4 and PFDN4 siRNA1/2 dose-dependent manner. (**F**,**G**) The protein-protein interaction between PFDN4 and CREBP1 was analyzed by CHIP in MCF7 and MDAMB231. (**H**,**I**) The promoter activation of AURKA was analyzed in 293T cells when pGL3 wild type (WT) and mutated (MT) fragments of the AURKA 5′-UTR and co-transfected with CREBP1 (**J**,**K**) Cell growth was analyzed in MCF7 and MDAMB231 when treated with 2 µg AURKA, 2 µg AURKA siRNA-1/2, 10 µM paclitaxel, 10 µM doxorubicin, 2 µg AURKA + 10 µM paclitaxel, 2 µg AURKA + 10 µM doxorubicin, 2 µg AURKA siRNA-1/2 + 10 µM paclitaxel and 2 µg AURKA siRNA-1/2 + 10 µM doxorubicin. *: *p* value < 0.05 was used to indicate a significant score.

## Data Availability

Data are contained within the article.

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
