# Peer review of "PFDN4 as a Prognostic Marker Was Associated with Chemotherapy Resistance through CREBP1/AURKA Pathway in Triple-Negative Breast Cancer"

_ijms, 2024, doi:10.3390/ijms25073906_

Round 1

Reviewer 1 Report

Comments and Suggestions for Authors

This study sheds light on the potential of PFDN4 in chemotherapy resistance and its molecular mechanism as therapeutic approaches for breast cancer. Building upon this insight, the authors extend their exploration to examine the applicability of PFDN4 in breast cancer treatment using publicly available data. They find that high PFDN4 expression is closely associated with breast invasive carcinoma. Elevated PFDN4 levels are also linked with advanced cancer stages and increased nodal metastasis status. Furthermore, their investigation also reveals PFDN4 has the ability to regulate resistance to doxorubicin and paclitaxel chemotherapy and its significant expression in triple negative breast cancer, but not non-triple negative breast cancer.

 One notable strength of the study is its comprehensive analysis using various data sources and methodologies to establish PFDN4's clinical relevance in breast cancer. The connections between PFDN4 and signaling pathways such as CREBP1 and AURKA signals, provide valuable insights into its potential role in breast cancer progression.

However, there are some aspects that could be further improved.

1.      Provide more detailed explanations of experimental methods and data analysis techniques used in breast cancer, such as UALCAN data and CRISPR knockout screening. This will enhance the readers' understanding and confidence in the results.

2.      The authors use MCF7 (non-triple negative) and MDAMB231 (triple negative) breast cancer cell lines to perform the effect of PFDN4 on cell viability and chemotherapy resistance experiments. However, the authors use different model in different cell lines throughout study: PFDN4 overexpression model in MCF7 cells, but PFDN4 knockdown model in MDAMB231 cells. Authors should make an effort to illustrate the inconsistency study design in different cell lines.

In conclusion, the findings presented in this study highlight the potential of PFDN4 as a predictive biomarker and therapeutic target in breast cancer contexts. The study effectively draws attention to the clinical implications of PFDN4 expression in breast cancer outcomes. Further research exploring the molecular mechanisms underlying PFDN4's involvement in breast cancer progression would be valuable to strengthen these findings.

Comments on the Quality of English Language

None

Author Response

Reviewer 1

This study sheds light on the potential of PFDN4 in chemotherapy resistance and its molecular mechanism as therapeutic approaches for breast cancer. Building upon this insight, the authors extend their exploration to examine the applicability of PFDN4 in breast cancer treatment using publicly available data. They find that high PFDN4 expression is closely associated with breast invasive carcinoma. Elevated PFDN4 levels are also linked with advanced cancer stages and increased nodal metastasis status. Furthermore, their investigation also reveals PFDN4 has the ability to regulate resistance to doxorubicin and paclitaxel chemotherapy and its significant expression in triple negative breast cancer, but not non-triple negative breast cancer.

One notable strength of the study is its comprehensive analysis using various data sources and methodologies to establish PFDN4's clinical relevance in breast cancer. The connections between PFDN4 and signaling pathways such as CREBP1 and AURKA signals, provide valuable insights into its potential role in breast cancer progression.

However, there are some aspects that could be further improved.

  1. Provide more detailed explanations of experimental methods and data analysis techniques used in breast cancer, such as UALCAN data and CRISPR knockout screening. This will enhance the readers' understanding and confidence in the results.

Response: Following the Reviewer’s comment, we have made the correction accordingly in the revision manuscript.” UALCAN is a comprehensive web-based resource for analyzing extensive cancer data, providing users with graphs and plots of gene performance and survival curves, as well as evaluating promoter DNA methylation information and performing pan-carcinogenic gene performance analyses[30]. In this study, we used UALCAN data to investigate the role of the PFDN4 gene among 24 human cancer subtypes. In addition, we also used UALCAN platform to analyze the correlation between PFDN4 expression and breast cancer patients with different clinicopathological characteristics. The expression of PFDN4 gene was measured as transcripts per million (TPM) reads and expression differences between the medians of the normal and cancer groups were compared by Student's t test. P value <0.05 was used to indicate a significant score.”

”Gene effect scores were released through Broad's Achilles and Sanger's SCORE projects, which are mainly big data values calculated through CRISPR knockout screening and combined with cancer genome ecology calculations. Negative scores represent cell growth inhibition and/or death after gene knockout, indicating that the gene may be required in a specific cell line. A score of 0 or above means that the gene is not an essential gene[32,33].”

2.The authors use MCF7 (non-triple negative) and MDAMB231 (triple negative) breast cancer cell lines to perform the effect of PFDN4 on cell viability and chemotherapy resistance experiments. However, the authors use different model in different cell lines throughout study: PFDN4 overexpression model in MCF7 cells, but PFDN4 knockdown model in MDAMB231 cells. Authors should make an effort to illustrate the inconsistency study design in different cell lines.

Response: Following the Reviewer’s comment, we have made the correction accordingly in the revision manuscript.” We also examined the expression of PFDN4 in breast cancer based on breast cancer subclasses including luminal, HER2 positive and triple negative group. The results showed that the expression of PFDN4 was significantly associated with triple negative compared to normal (Figure 2A, P<0.01). We also used CRISPR knockout screens of the PFDN4 system to analyze cell growth in 12 non-triple negative and 18 triple negative breast cancer cell lines. The results showed that silencing PFDN4 has a greater ability to inhibit cell growth in triple-negative compared to non-triple-negative breast cancer cell lines (Figure 2B and C, P=0.039). We also examined the cell cycle by flow cytometry in the breast cancer cell lines MCF7 (non-triple negative) and MDAMB231 (triple negative). The results showed that PFDN4 overexpression decreased the percentage of cells in G1/M phase, but increased S phase in MCF7 (Figure 2D). In contrast, PFDN4 knockdown decreased the percentage of cells in S phase (Figure 2E).”, “In addition, the results showed that the level of PFDN4 in MDAMB231 cells is higher than that in MCF7 (Figure 4 A). We also used a PFDN4 overexpression plasmid and siRNA-1/2 to detect cell growth by CCK-8 assay when doxorubicin and paclitaxel treatment in MCF7 and MDAMB231 cell lines.” Therefore, we selected MCF7 (non-triple negative) and MDAMB231 (triple negative) breast cancer cell lines for related experiments.”

In conclusion, the findings presented in this study highlight the potential of PFDN4 as a predictive biomarker and therapeutic target in breast cancer contexts. The study effectively draws attention to the clinical implications of PFDN4 expression in breast cancer outcomes. Further research exploring the molecular mechanisms underlying PFDN4's involvement in breast cancer progression would be valuable to strengthen these findings.

Response: Thanks to the reviewer's comments, in the future we will further study and explore the molecular mechanism of PFDN4's involvement in breast cancer progression, which will be helpful for the treatment of breast cancer.

Reviewer 2 Report

Comments and Suggestions for Authors

The manuscript reports on the possible role of FDN4 as a key player in chemoresistance in triple-negative breast cancer. High expression of FDN4 appears to carry worse disease-specific survival and chemoresistance to taxanes and doxorubicin, as assessed in vitro

In Introduction it is not quite clear why the focus of the research was on the 4 beta subunits PFDN 1, 2, 4 and 6 while the alpha subunit were not investigated; the authors are advised to describe in more details the rationale behind their research

Also, high expression of PFDN4 seems to hinder antitumor efficacy of chemotherapeutic agents belonging to different families and with different mechanisms of action, i.e. paclitaxel and docetaxel (mitotic spindle agents), cyclophosphamide (alkylating agents), doxorubicin (antitumor antibiotics). The authors are invited to discuss these findings in more details in Discussion

Last, from a translational medicine standpoint, the authors ideally should discuss

-          how many subjects with breast cancer display high expression of PFDN4 thus potentially benefiting from treatment with PFDN4 inhibitors, and

-          what the untoward effects, if any, of inhibition of PFDN4 could be based upon its function in normal tissues, and if selective inhibition of PFDN4 versus the remaining alpha and beta subunit is a requirement  

MINOR COMMENTS

Title

what does "acted" mean?

Abstract

line 18: breast cancer is not an endocrine malignancy

line 19: please rephrase the sentences beginning with “Although” and “However”, respectively

line 24: “statistically significant” should be “statistically significantly associated”

line 24: what does “subclasses” mean?

line 24: should “triple negative” be “triple-negative subtype”?

line 24: should “disease-specific” be “disease-specific survival”?

line 26: please consider replacing “had” with “exhibited” and “better” with “stronger”

line 26: where in the manuscript body is information on 41/46 cell lines inhibited?

Introduction

line 37: please consider neoadjuvant chemotherapy as a treatment option in breast cancer

line 39: paclitaxel binds to microtubules and cause assembly of nonfunctional microtubules but it does not cause double strand breaks in DNA

line 57: CREBP1 is not described in its function

Results

line 62: please consider removing “human” and changing “patients” into “subjects”

figure 1: please label Y axis as “disease-specific survival”

figure 1: please increase the font of X axis legend

figure 1: please discuss the possible impact on statistics of disparate numbers of high and low expression in panels A-D

figure 1 panel G: is expression of PFDN4 higher in normal tissue than in stage 1 breast cancer? Is normal tissue normal breast tissue?

figure 1 panel H: is expression of PFDN4 higher in normal tissue than in N3? Is normal tissue normal breast tissue?

figure 2 panel D: it is very difficult to match different shades of gray with phases of cell cycle; that said, there appears to be no difference between control and PFDN4 in the percentage of cells in G2/M

figure 2 panel E: it is very difficult to match different shades of gray with phases of cell cycle

figure 3: title of the figure should be in bold in the legend (lines 101-102)

figure 3: please label Y axis as “disease-specific survival”

figure 3: please discuss the possible impact on statistics of disparate numbers of high and low expression in panels A-F

figure 3 panel F: there is no comment on results with docetaxel, which seems to be in keeping with those obtained with paclitaxel (the two drugs share the same mechanism of action)

lines 106-107 please rephrase

line 110: what is cyclitaxel?

figure 4: please rephrase the legend for panels B, C and for panels D, E (lines 119-122)

figure 5: should “CRRBP1” be “CREBP1” (line 147)?

figure 5: please rephrase the legend to panel D, E (lines 147-149)

figure 5: please rephrase the legend to panels H, I (lines 150-152)

lines 172 and 173: please change “toxicity” into “antitumor activity”

Materials and Methods

line 218: please specify the subtype of breast cancer for both MCF7 and MDAMB231

Comments on the Quality of English Language

This reviewer is not mother tongue, however, English can be improved in selected sections of the text, indicated in my comments

Author Response

Reviewer 2

The manuscript reports on the possible role of FDN4 as a key player in chemoresistance in triple-negative breast cancer. High expression of FDN4 appears to carry worse disease-specific survival and chemoresistance to taxanes and doxorubicin, as assessed in vitro

In Introduction it is not quite clear why the focus of the research was on the 4 beta subunits PFDN 1, 2, 4 and 6 while the alpha subunit were not investigated; the authors are advised to describe in more details the rationale behind their research

Response: Reviewer’s comment is very well, we add the section in the revision manuscript.” Prefolding protein subunit (PFDN) consists of six different subunits (PFDN1-6), two α subunits (PFDN3 and 5) and four β subunits (PFDN1, 2, 4 and 6)[4] and is a hexameric cofacilitator complex that helps regulate the monomeric folding of actin and tubulin[5]. Past studies have found that all six PFDN subunits are involved in cancer-related biology[6-9]. Among them, PFDN3 and PFDN5 subunits have been shown to form complexes with many proteins[10,11]. However, there is still much to be clarified in the study of β subunits. The expression of FDN4, also known as C-1, is a transcription factor and has the ability to regulate the cell cycle[12,13] and may be closely associated with the occurrence, development and poor prognosis of several tumors, including hepatocellular carcinoma and colorectal cancer[14].”

Also, high expression of PFDN4 seems to hinder antitumor efficacy of chemotherapeutic agents belonging to different families and with different mechanisms of action, i.e. paclitaxel and docetaxel (mitotic spindle agents), cyclophosphamide (alkylating agents), doxorubicin (antitumor antibiotics). The authors are invited to discuss these findings in more details in Discussion

Response: Reviewer’s comment is very well, we add the section in the revision manuscript. “The PFDN family has also been linked to drug resistance in previous studies. PFDN2 has the ability to regulate the folding of microtubule proteins, which when mutated or abnormal can lead to misfolding of microtubule proteins, and Taxel can inhibit cell division and slow the proliferation of cancer cells by stabilizing microtubules and destroying the mitotic spindle[24]. Therefore, PFDN2 has the ability to regulate the development of resistance to Taxel treatment in breast cancer. Our PFDN4 has the ability to antagonize mitotic spindle agents (paclitaxel), so we suggest that PFDN4 has the ability to affect mitotic spindle. In the future, we will further explore how PFDN4 regulates mitotic spindle in the breast cancer. Cyclophosphamide is an Alkylating agent that can alkylate DNA guanine and form bonds with double-stranded DNA, making cellular DNA irreparable[25]. Doxorubicin also has the ability to rapidly inhibit mitotic activity and inhibit DNA nucleic acid synthesis[26]. In our study, PFDN4 showed higher expression in breast cancer than in normal tissue and in triple-negative breast cancer than in non-triple-negative breast cancer, and we found that PFDN4 increases the S phase of cell cycle and affects the activity of the drug through mediated cell cycles, which provides an effective mechanism for verifying drug resistance and novelty.”

Last, from a translational medicine standpoint, the authors ideally should discuss

 how many subjects with breast cancer display high expression of PFDN4 thus potentially benefiting from treatment with PFDN4 inhibitors, and what the untoward effects, if any, of inhibition of PFDN4 could be based upon its function in normal tissues, and if selective inhibition of PFDN4 versus the remaining alpha and beta subunit is a requirement

Response: Reviewer’s comment is very well, we add the section in the revision manuscript. “Currently, about 5.6% (60/1003 (Low expression) + 60 (High expression)) of breast cancer patients are from the PFDN4 high expression group. In our study suggested that PFDN4 high-expressing groups should reduce the use of mitotic spindle, alkylating agents for drug treatment. Then use Tamoxifen or Anastrazole drugs to improve the healing effect. In addition, PFDN4 inhibitors or other relatively α- and β-subunit-selective PFDN inhibitors can also be used in future clinical drug development.”

MINOR COMMENTS

Title

what does "acted" mean?

Response: Following the Reviewer’s comment, we have made the correction accordingly in the revision manuscript.” FDN4 as a prognostic marker was associated with chemotherapy resistance through CREBP1/AURKA pathway in triple-negative breast cancer”

Abstract

line 18: breast cancer is not an endocrine malignancy

Response: Following the Reviewer’s comment, we have made the correction accordingly in the revision manuscript.” Breast cancer is the most common malignancy and its incidence is increasing.”

line 19: please rephrase the sentences beginning with “Although” and “However”, respectively

Response: Following the Reviewer’s comment, we have made the correction accordingly in the revision manuscript.” It is currently mainly treated by clinical chemotherapy, but chemoresistance remains poorly understood.”

line 24: “statistically significant” should be “statistically significantly associated”

Response: Following the Reviewer’s comment, we have made the correction accordingly in the revision manuscript.” Our study found that PFDN4 was highly expressed in breast cancer compared to normal tissues and was statistically significantly associated with stage, nodal status, subclasses(luminal, HER2 positive and triple negative), triple-negative subtype and disease-specific survival by TCGA database analysis.”

line 24: what does “subclasses” mean?

Response: Following the Reviewer’s comment, we have made the correction accordingly in the revision manuscript. “subclasses(luminal, HER2 positive and triple negative)”

line 24: should “triple negative” be “triple-negative subtype”?

Response: Following the Reviewer’s comment, we have made the correction accordingly in the revision manuscript.” triple-negative subtype and disease-specific survival by TCGA database analysis.”

line 24: should “disease-specific” be “disease-specific survival”?

Response: Following the Reviewer’s comment, we have made the correction accordingly in the revision manuscript.” ” triple-negative subtype and disease-specific survival by TCGA database analysis.”

line 26: please consider replacing “had” with “exhibited” and “better” with “stronger”

Response: Following the Reviewer’s comment, we have made the correction accordingly in the revision manuscript.” CRISPR knockout PFDN4 inhibited the growth of 89% of breast cancer cell lines, and the triple-negative cell line exhibited a stronger inhibitory effect than the non-triple-negative cell line.”

line 26: where in the manuscript body is information on 41/46 cell lines inhibited?

Response: Following the Reviewer’s comment, we have made the correction accordingly in the revision manuscript.” CRISPR knockout PFDN4 inhibited the growth of 89% of breast cancer cell lines,”

Introduction

line 37: please consider neoadjuvant chemotherapy as a treatment option in breast cancer

Response: Following the Reviewer’s comment, we have made the correction accordingly in the revision manuscript. “Currently, neoadjuvant chemotherapy as a treatment option in breast cancer including anthracyclines and paclitaxel are used as systemic primary therapies in chemotherapy.”

line 39: paclitaxel binds to microtubules and cause assembly of nonfunctional microtubules but it does not cause double strand breaks in DNA

Response: Following the Reviewer’s comment, we have made the correction accordingly in the revision manuscript. “The chemotherapeutic effect of anthracyclines is to induce DNA double-strand breaks (DSBs), leading to apoptosis[1] and paclitaxel binds to microtubules and cause assembly of nonfunctional microtubules.”

line 57: CREBP1 is not described in its function

Response: Following the Reviewer’s comment, we have made the correction accordingly in the revision manuscript. “Previous study found that PFDN4 binds to CREBP1[11] and transcription factor affinity prediction (TRAP) shows that CREBP1 is an upstream transcription factor of AURKA (Figure 5C). CREB is a basic region transcription factor because it possesses a leucine zipper domain(bZIP) that promotes dimerization and mediates its binding to DNA. Among them, CREBP1 (also known as ATF-2) is a highly homologous protein and also has a bZIP structure.”

Results

line 62: please consider removing “human” and changing “patients” into “subjects”

Response: Following the Reviewer’s comment, we have made the correction accordingly in the revision manuscript.” Association of PFDN with disease-specific survival and clinicopathological features in breast cancer subjects”

figure 1: please label Y axis as “disease-specific survival”

Response: Following the Reviewer’s comment, we have made the correction accordingly in the revision manuscript figure 1.” disease-specific survival”

figure 1: please increase the font of X axis legend

Response: Following the Reviewer’s comment, we have made the correction accordingly in the revision manuscript figure 1. ”Months”

figure 1: please discuss the possible impact on statistics of disparate numbers of high and low expression in panels A-D

Response: Following the Reviewer’s comment, we have made the correction accordingly in the revision manuscript. ”First, we evaluated the prognostic value of disease-specific with PFDN family including PFDN1, 2, 4 and 6 by TCGA PanCancer Atlas using Kaplan-Meier analysis of cBioPortal database(https://www.cbioportal.org) in breast invasive carcinoma (Figure 1 A, B, C and D). The PFDN4 gene samples were divided into two groups using the median performance of the gene (high vs. low expression). The results show that high expression of PFDN4 was significantly associated with survival rate (P=0.0235) in breast invasive carcinoma.”

figure 1 panel G: is expression of PFDN4 higher in normal tissue than in stage 1 breast cancer? Is normal tissue normal breast tissue?

Response: Following the Reviewer’s comment, we have made the correction accordingly in the revision manuscript. This study use the median of PFDN4 for analysis in normal and cancer groups. “In this study, we used UALCAN data to investigate the role of the PFDN4 gene among 24 human cancer subtypes. In addition, we also used UALCAN platform to analyze the correlation between PFDN4 expression and breast cancer patients with different clinicopathological characteristics. The expression of PFDN4 gene was measured as transcripts per million (TPM) reads and expression differences between the medians of the normal and cancer groups were compared by Student's t test.”

figure 1 panel H: is expression of PFDN4 higher in normal tissue than in N3? Is normal tissue normal breast tissue? 

Response: Following the Reviewer’s comment, we have made the correction accordingly in the revision manuscript. This study use the median of PFDN4 for analysis in normal and cancer groups. “In this study, we used UALCAN data to investigate the role of the PFDN4 gene among 24 human cancer subtypes. In addition, we also used UALCAN platform to analyze the correlation between PFDN4 expression and breast cancer patients with different clinicopathological characteristics. The expression of PFDN4 gene was measured as transcripts per million (TPM) reads and expression differences between the medians of the normal and cancer groups were compared by Student's t test.”

figure 2 panel D: it is very difficult to match different shades of gray with phases of cell cycle; that said, there appears to be no difference between control and PFDN4 in the percentage of cells in G2/M

Response: Following the Reviewer’s comment, we have made the correction accordingly in the revision manuscript figure 2D and add this section in the results” MCF7 and MDAMB231 were no difference between control and PFDN4 in the percentage of cells in G2/M”.

figure 2 panel E: it is very difficult to match different shades of gray with phases of cell cycle

Response: Following the Reviewer’s comment, we have made the correction accordingly in the revision manuscript figure 2E

figure 3: title of the figure should be in bold in the legend (lines 101-102)

Response: Following the Reviewer’s comment, we have made the correction accordingly in the revision manuscript. “Figure. 3 PFDN4 is a chemotherapy resistance factor associated with survival when treated with chemotherapeutic drugs in TCGA breast cancer.”

figure 3: please label Y axis as “disease-specific survival”

Response: Following the Reviewer’s comment, we have made the correction accordingly in the revision manuscript figure 3. “disease-specific survival”

figure 3: please discuss the possible impact on statistics of disparate numbers of high and low expression in panels A-F

Response: Following the Reviewer’s comment, we have made the correction accordingly in the revision manuscript. ” The prognostic value of disease-specific of PFDN4 (PFDN4 uses median to differentiate between high and low expression) with drug including doxorubicin (Fig. 3A), paclitaxel (Fig. 3B), cyclophosphamide (Fig. 3C), tamoxifen (Fig. 3D), anastrazole (Fig. 3E) and docetaxel (Fig. 3F) were determined using Kaplan-Meier analysis in breast invasive carcinoma. The results showed that high expression of PFDN4 with doxorubicin (P=0.0003701), paclitaxel (P=0.001294) and cyclophosphamide (P=0.003080) were signifi-cantly associated with survival rate in breast invasive carcinoma, but the other drug was not. Doxorubicin, paclitaxel and cyclophosphamide do not improve survival in breast cancer patients with high PFDN4 expression. These results suggest that PFDN4 has an antagonistic effect on doxorubicin, paclitaxel and cyclophosphamide in invasive breast cancer.”

figure 3 panel F: there is no comment on results with docetaxel, which seems to be in keeping with those obtained with paclitaxel (the two drugs share the same mechanism of action)

Response: Following the Reviewer’s comment, because docetaxel was not statistically significant in survival analysis with PFDN4, we did not conduct a further discussion on docetaxel.

lines 106-107 please rephrase

Response: Following the Reviewer’s comment, we have made the correction accordingly in the revision manuscript. “The prognostic value of disease-specific of PFDN4 (PFDN4 uses median to differentiate between high and low expression) with drug including doxorubicin (Fig. 3A), paclitaxel (Fig. 3B), cyclophosphamide (Fig. 3C), tamoxifen (Fig. 3D), anastrazole (Fig. 3E) and docetaxel (Fig. 3F) were determined using Kaplan-Meier analysis in breast invasive carcinoma, respectively.”

line 110: what is cyclitaxel?

Response: Following the Reviewer’s comment, we have made the correction accordingly in the revision manuscript. “paclitaxel”

figure 4: please rephrase the legend for panels B, C and for panels D, E (lines 119-122)

Response: Following the Reviewer’s comment, we have made the correction accordingly in the revision manuscript “The cell growth was analyzed by dose-dependent manner in PFDN4 plasmid(B), siRNA-1/2(C), doxorubicin(D) and paclitaxel(E) in MCF7 and MDAMB231, respectively.”

figure 5: should “CRRBP1” be “CREBP1” (line 147)?

Response: Following the Reviewer’s comment, we have made the correction accordingly in the revision manuscript. “CREBP1”

figure 5: please rephrase the legend to panel D, E (lines 147-149)

Response: Following the Reviewer’s comment, we have made the correction accordingly in the revision manuscript. “The expression of AURKA was analyzed in MCF7 and MDAMB231 when dose-dependent manner in PFDN4 and PFDN4 siRNA1/2.”

figure 5: please rephrase the legend to panels H, I (lines 150-152)

Response: Following the Reviewer’s comment, we have made the correction accordingly in the revision manuscript. “The promoter activation of AURKA was analyzed in 293T cells when pGL3 wild type (WT) and mutated (MT) fragments of the AURKA 5′-UTR and cotransfected with CREBP1”

lines 172 and 173: please change “toxicity” into “antitumor activity”

Response: Following the Reviewer’s comment, we have made the correction accordingly in the revision manuscript. “AURKA overexpression could antagonize the toxicity of doxorubicin and paclitaxel drugs, and AURKA siRNA 1/2 could enhance the antitumor activity of paclitaxel and doxoru-bicin drugs in human breast cancer cell lines (Figure 5 J, K).”

Materials and Methods

line 218: please specify the subtype of breast cancer for both MCF7 and MDAMB231

Response: Following the Reviewer’s comment, we have made the correction accordingly in the revision manuscript. “Human breast cancer cell lines (MCF7, non-triple negative and MDAMB231, triple negative) were purchased from the American Type Culture Collection (ATCC)”

Reviewer 3 Report

Comments and Suggestions for Authors

In this manuscript the authors investigate could FDN4 be used as prognostic marker for chemotherapy resistance in breast cancer through CREBP1/AURKA pathway. This topic is interesting and could add to the general knowledge about breast cancer therapy. I have some concerns about this study which are numbered here point by point:

1.       The results section is poorly written, and a proper description of the results should be provided. For example, the authors state that their results show that high expression of PFDN4 is associated with cancer stage, but don’t comment which stage, how is it associated and is it statistically significant. Another example is figure 1E which is barely commented on in the results. Why did they add this graph? What does it say in the author’s opinion? This pattern is visible across the results section.

2.       Figure 1 graphs are too small and impossible to read. Please improve these graphs.

3.       Figure 5 graphs D-K don’t have statistical data. Are any of the differences the authors observed statistically significant?

4.       In the discussion section the authors describe their results and propose possible meaning. They should try to find and include work by other authors and compare possible differences or similarities and their implications.

5.       In the section 4.3. Cell growth, can the authors clarify which microplate reader was used for the experiments.

6.       In sections 4.5, 4.6, and 4.7 the authors should specify in more detail the inquiry/parameters/methodology that was used.

Comments on the Quality of English Language

    There are language errors throughout the manuscript which should be corrected by a native English speaker.

Author Response

Reviewer 3

In this manuscript the authors investigate could FDN4 be used as prognostic marker for chemotherapy resistance in breast cancer through CREBP1/AURKA pathway. This topic is interesting and could add to the general knowledge about breast cancer therapy. I have some concerns about this study which are numbered here point by point:

  1. The results section is poorly written, and a proper description of the results should be provided. For example, the authors state that their results show that high expression of PFDN4 is associated with cancer stage, but don’t comment which stage, how is it associated and is it statistically significant. Another example is figure 1E which is barely commented on in the results. Why did they add this graph? What does it say in the author’s opinion? This pattern is visible across the results section.

Response: Following the Reviewer’s comment, we have made the correction accordingly in the revision manuscript. “Next, the expression of PFDN4 across TCGA pan-cancer was shown in Figure 1E(BRCA: Breast invasive carcinoma) found that PFDN4 is highly expressed in most tumors relative to normal tissue, and the result also indicated that the expression level of PFDN4 was higher in BRCA (n=1097) compared to its matched normal tissues(n=114)(Figure 1F) through UALCAN database(https://ualcan.path.uab.edu). In addition, high expression of PFDN4 was associated with stage1, 2 and 3 (Fig. 1G) and nodal metastasis status (N0, N1 and N2) (Fig. 1H). This results found that high expression of PFDN4 was a prognostic factor and associated with tumor stage and nodal metastasis.”

  1. Figure 1 graphs are too small and impossible to read. Please improve these graphs.

Response: Following the Reviewer’s comment, we have made the correction accordingly in the revision manuscript Figure 1.

  1. Figure 5 graphs D-K don’t have statistical data. Are any of the differences the authors observed statistically significant?

Response: Following the Reviewer’s comment, we have made the correction accordingly in the revision manuscript. We've plotted the statistics on the chart.

  1. In the discussion section the authors describe their results and propose possible meaning. They should try to find and include work by other authors and compare possible differences or similarities and their implications.

Response: Following the Reviewer’s comment, we add the section in the revision manuscript. “Past studies have also found that PFDN4 is highly expressed in breast cancer cell lines and identified as an oncogenic gene[12]. This is consistent with our current research, and our research further found that PFDN4 has the ability to antagonize chemotherapy drugs in breast cancer. In addition, a study conducted by Miyoshi et al. in colorectal cancer found that high expression of PFDN4 may be an indicator of relatively well prognosis[14]. These findings suggest that PFDN4 may play different roles in different tumors.”

  1. In the section 4.3. Cell growth, can the authors clarify which microplate reader was used for the experiments.

Response: Following the Reviewer’s comment, we have made the correction accordingly in the revision manuscript. “5X103 cells were seeded in 96-well plates after transfection and incubated 24 hours later by cell growth assay using Cell Count Kit-8 (CCK-8, Dojindo, Kumamoto, Japan). Absorbance (450 nm) was detected in each sample using a microplate reader(Bio-Rad, Hercules, CA, USA).”

  1. In sections 4.5, 4.6, and 4.7 the authors should specify in more detail the inquiry/parameters/methodology that was used.

Response: Following the Reviewer’s comment, we have made the correction accordingly in the revision manuscript.

4.5. UALCAN

UALCAN is a comprehensive web-based resource for analyzing extensive cancer data, providing users with graphs and plots of gene performance and survival curves, as well as evaluating promoter DNA methylation information and performing pan-carcinogenic gene performance analyses[30]. In this study, we used UALCAN data to investigate the role of the PFDN4 gene among 24 human cancer subtypes. In addition, we also used UALCAN platform to analyze the correlation between PFDN4 expression and breast cancer patients with different clinicopathological characteristics. The expression of PFDN4 gene was measured as transcripts per million (TPM) reads and expression dif-ferences between the medians of the normal and cancer groups were compared by Student's t test. P value <0.05 was used to indicate a significant score.

4.6. cBioPortal

The cBioPortal Cancer Genome Database is an open-access web platform for inter-active exploration of cancer genomic data among many cancer genomes[31]. We selected cancer study "Breast Invasive Carcinoma (TCGA, PanCancer Atlas)" and  PFDN4 for our study as we obtained data from an open access database. We analysed the survival rate of PFDN4 in breast cancer and compared the survival rate after clinical drug treatment.

4.7. Gene Effect Scores

Gene effect scores were released through Broad's Achilles and Sanger's SCORE projects, which are mainly big data values calculated through CRISPR knockout screening and combined with cancer genome ecology calculations. Negative scores represent cell growth inhibition and/or death after gene knockout, indicating that the gene may be required in a specific cell line. A score of 0 or above means that the gene is not an essential gene[32,33].

Round 2

Reviewer 1 Report

Comments and Suggestions for Authors

The authors have answered all my questions and made the necessary changes to the manuscript.

Comments on the Quality of English Language

The present language quality is not good enough and needs to be improved, especially in the Discussion parts.

Reviewer 2 Report

Comments and Suggestions for Authors

The authors have addressed reviewer's comment

However, English is difficult to read here and there